# Genetics, Genomics and Emerging Molecular Therapies of Pancreatic Cancer

**DOI:** 10.3390/cancers15030779

**Published:** 2023-01-27

**Authors:** Jakub Liu, Magdalena Mroczek, Anna Mach, Maria Stępień, Angelika Aplas, Bartosz Pronobis-Szczylik, Szymon Bukowski, Magda Mielczarek, Ewelina Gajewska, Piotr Topolski, Zbigniew J. Król, Joanna Szyda, Paula Dobosz

**Affiliations:** 1Biostatistics Group, Wroclaw University of Environmental and Life Sciences, 51-631 Wroclaw, Poland; 2Centre for Cardiovascular Genetics and Gene Diagnostics, Foundation for People with Rare Diseases, Wagistrasse 25, 8952 Schlieren, Switzerland; 3Department of Psychiatry, Medical University of Warsaw, 00-665 Warsaw, Poland; 4Central Clinical Hospital of Ministry of the Interior and Administration in Warsaw, 02-507 Warsaw, Poland; 5Department of Infectious Diseases, Doctoral School, Medical University of Lublin, 20-059 Lublin, Poland; 6National Research Institute of Animal Production, Krakowska 1, 32-083 Balice, Poland

**Keywords:** pancreatic cancer, genetics, genomics, molecular therapies

## Abstract

**Simple Summary:**

This review article is a summary of the current state of knowledge regarding the genetics of pancreatic cancer and a presentation of possible treatment options reflecting genomic medicine advances and a personalised medicine approach. Several oncogenes in which somatic changes lead to the development of tumours, including genes BRCA1/2 and PALB2, TP53, CDKN2A, SMAD4, MLL3, TGFBR2, ARID1A and SF3B1, are involved in pancreatic cancer. Between 4% and 10% of individuals with pancreatic cancer will have a mutation in one of these genes. Six percent of patients with pancreatic cancer have NTRK pathogenic fusion. It is estimated that from 24% to as many as 44% of pancreatic cancers show homologous recombination deficiency (HRD). The most common cause of HRD are inactivating mutations in the genes regulating this DNA repair system, mainly BRCA1 and BRCA2, but also PALB2, RAD51C and several dozen others.

**Abstract:**

The number of cases of pancreatic cancers in 2019 in Poland was 3852 (approx. 2% of all cancers). The course of the disease is very fast, and the average survival time from the diagnosis is 6 months. Only <2% of patients live for 5 years from the diagnosis, 8% live for 2 years, and almost half live for only about 3 months. A family predisposition to pancreatic cancer occurs in about 10% of cases. Several oncogenes in which somatic changes lead to the development of tumours, including genes BRCA1/2 and PALB2, TP53, CDKN2A, SMAD4, MLL3, TGFBR2, ARID1A and SF3B1, are involved in pancreatic cancer. Between 4% and 10% of individuals with pancreatic cancer will have a mutation in one of these genes. Six percent of patients with pancreatic cancer have NTRK pathogenic fusion. The pathogenesis of pancreatic cancer can in many cases be characterised by homologous recombination deficiency (HRD)—cell inability to effectively repair DNA. It is estimated that from 24% to as many as 44% of pancreatic cancers show HRD. The most common cause of HRD are inactivating mutations in the genes regulating this DNA repair system, mainly BRCA1 and BRCA2, but also PALB2, RAD51C and several dozen others.

## 1. Introduction

Cancer remains an escalating problem worldwide. The reflection of this disturbing trend can be also noticed in Poland and backed up by the statistical data. The number of cases of pancreatic cancers (PC) in 2019 was 3852 (approx. 2% of all cancers) [1]. In 2019, 100,324 people died from cancer in Poland, including 5068 people from pancreatic cancer [1]. The course of the disease is very fast, and the average survival time from the diagnosis is 6 months [2]. Only <2% of patients live for 5 years from the diagnosis, 8% live for 2 years, and almost half live for only about 3 months [2,3]. In the case of resectable tumours, the 5-year survival rate is approximately 20%, but only 10–20% of patients are diagnosed at such a stage of disease that tumour resection is a possible option [4]. Most cases (approx. 90%) of pancreatic cancer occur after the age of 50. The risk of pancreatic cancer increases with age, reaching its peak in the ninth percentile in both sexes (45–50/100,000 inhabitants) [5]. Cases < 40 years of age are extremely rare, although globally, there is an increase in this age group—which is usually associated with worse prognosis [2]. Existing studies indicate that the progression of the disease from the initiation to the formation of metastatic pancreatic cancer takes up to 15 years [6]. This creates enough time to recognize the disease early. However, the diagnosis of pancreatic cancer at an early stage, despite recent advances in diagnostic methods, is still very difficult, resulting in the delayed implementation of a potentially effective treatment. Pancreatic cancer patient survival has only slightly improved over the past two decades [7]. At the time of diagnosis, up to 80% of patients already have advanced, metastatic disease, which precludes the effective use of surgical treatment and many other modern therapeutic methods [2]. 

There are two main types of pancreatic cancer: non-endocrine pancreatic cancer and neuroendocrine pancreatic cancer [8]. The non-endocrine type is divided into four subtypes. The most common of these is adenocarcinoma, which is the cause of roughly 90% of pancreatic cancer diagnoses. This type of cancer emerges in the lining of the pancreatic ducts. Rarely, it can also have its origin in the cells that produce the pancreatic enzymes. Another subtype of the non-endocrine cancer type is colloid carcinoma, which accounts for 1–3% of non-endocrine pancreatic cancers [8]. This subtype has its origin in the intraductal papillary mucinous neoplasm. Colloid carcinoma is easier to treat than other pancreatic cancer types [9].

Adenosquamous carcinoma is the next type of non-endocrine pancreatic cancer. It is the most aggressive type and accounts for 1–4% of diagnoses. The rarest type of pancreatic cancer is squamous cell carcinoma, which is also the least studied subtype, but studies suggest that it has its origins in the pancreatic ducts. The second large class of pancreatic cancers are neuroendocrine pancreatic cancers. This type is rarer than the non-endocrine class, making up less than 5% of cases. This class of cancers develops from the pancreatic gland that releases pancreatic hormones (glucagon and insulin) into the bloodstream [8,9].

On the other hand, the possibility of targeted treatment is the major motivation for the identification of genetic biomarkers in pancreatic cancers. Several genetic biomarkers have proven to be associated with a treatment outcome, among them variants in the following genes: *K-ras*, *hENT1*, *DCK*, *CDA*, *DPD*, *HMLH1*, *TP1DPD*, *HER2* and *SMAD4* [10]. Treatment options depend on the molecular profile of the tumour. For example, tumours with *NTRK* fusion may be treated with a larotrectinib; olaparib treatment targets a mutation in the *BRCA1* or *BRCA2* gene, and pembrolizumab is approved by FDA only in the cancers with MSI-H (microsatellite instability high) or MMR-D (mismatch repair deficiency). 

Most of the clinical trials that involve molecular profiles of the pancreas tumour are those including patients with germline *BRCA1*/*BRCA2* pathogenic variants. Most of these involve PARPi, either as a single treatment, together with other oncostatics or as a maintenance therapy. However, in an early stage, there are promising therapies targeting other molecular pathways. For example, there are ongoing trials to implement RAS-directed therapies in pancreatic cancer treatment. One of such implementations is neoantigen T-cell receptor gene therapy, in which a single infusion of genetically engineered autologous T-cells that target mutant KRAS G12D driver mutation resulted in the regression of metastatic pancreatic cancer [11]. Some studies including immunotherapies, e.g., PD-1 immune checkpoints, are recruiting [12,13]. Though the search for a treatment is ongoing, there are some prominent possibilities for early detection. One of these is the correlation between new-onset diabetes and pancreatic cancer [14]. In a situation in which time is of the essence, this might be a new area to expand upon in early detection. Moreover, it has been shown that patients with diabetes mellitus (DM) have significantly higher median overall survival than those without DM [15].

Meanwhile, in many other cancers, patient mortality is systematically decreasing, which is related, inter alia, to the personalization of therapies and the use of targeted drugs. 

## 2. Genetics of Pancreatic Cancer 

Pancreatic cancer, like all cancers, is a genomic disease caused by abnormalities in the sequence or structure of the genomic DNA. Genetic changes leading to neoplasm can be initiated by family predispositions (germline mutations) or by acquired changes (somatic mutations), which occur, for example, because of chronic inflammation in the pancreas [6]. Due to the inaccessible location of many tumours, difficult sampling and the extremely short survival time of patients, the exact molecular basis of the formation of these neoplasms has not been well understood. It is estimated that a family predisposition to pancreatic cancer occurs in about 10% of cases [16,17]. There are also other genetic pathologies that may indirectly lead to the development of pancreatic neoplasms, including hereditary pancreatitis, during which recurring episodes of acute inflammation occur in childhood. There are also several other oncogenes in which somatic changes lead to the development of pancreatic tumours, including genes *BRCA1*, *BRCA2*, *PALB2*, *TP53*, *CDKN2A*, *SMAD4*, *MLL3*, *TGFBR2*, *ARID1A* and *SF3B1* [18,19,20]. There are also gene fusions, especially in the case of *NTRK*: it is estimated as many as 6% of patients with pancreatic cancer have *NTRK* pathogenic fusion [21]. Currently, the most promising targeted therapies for pancreatic cancer target tumours that are characterised by homologous recombination deficiency (HRD). The mechanism of homologous recombination is responsible for the repair of DNA double-strand breaks [22]. This damage leads to numerous genomic rearrangements and the so-called genomic instability [23]. It is estimated that from 24% to as many as 44% of pancreatic cancers show HRD [23]. The most common cause of HRD are inactivating mutations in the genes regulating this DNA repair system, mainly *BRCA1* and *BRCA2*, but also *PALB2*, *RAD51C* and several dozen others [22]. 

## 3. Genomics of Pancreatic Cancer

The mutation’s diversity can be studied thanks to recent advances in sequencing technologies (next-generation sequencing; NGS) and computational biology, which have provided the possibility of the reliable detection of molecular alterations, not only for single genes but on the whole genome scale [24]. According to Felsenstein et al., recent genomic studies, including whole exome and whole genome sequencing, have contributed to a better understanding of the complex genomic changes due to a combination of large-scaled structural variants (SV) and widespread single-nucleotide variants (SNV) [25]. As an example, SVs can affect tumour suppressors, including *SMAD4* and *CDKN2A,* by causing homozygous deletion or inactivating alterations [26]. Pancreatic ductal adenocarcinoma subtypes (stable, locally rearranged, scattered, or unstable) have been defined based on the associated SV’s features, such as the SVs’ count, the predominance of specific SV types and SV’s distribution across the genome of each patient [19]. Moreover, *BRCA1*, *BRCA2*, *PALB2* and other genes from the Fanconi anaemia (FA) pathway altered by structural germline or somatic mutations are associated with the unstable subtype, which in turn is caused by the inability to repair double-stranded DNA breaks [27]. In addition, Zhang et al. detected recurrent somatic mutations in multiple genes, including *KRAS*, *TP53*, *CDKN2A*, *SMAD4*, *ARID1A* and *CDKN2B,* in 1080 Chinese patients with pancreatic ductal adenocarcinoma, demonstrating the genomic characteristics on a large scale [28].

Several SNVs have been found to be associated with pancreatic cancer. Some of the most common mutations are the so-called BRCA1 and BRCA2 mutations (as shown in Table 1). In Table 1, the polymorphisms in the PALB2 gene are shown, which have also been found to be associated. Similarly, the mutations in the CDKN2A, KRAS and genes that are related to PC are summarised in Table 1, as well as mutations in the GNAS, ATM, MLH1, MSH2, MSH6 and PSM2 genes (found to cause pancreatic cancer directly or indirectly). Mutations in the genes encoding the CPA1 and CPB1 carboxypeptidases have also been found to be causal (shown in Table 1).

Currently, one of the biggest issues in pancreatic cancer treatment is the considerable genetic heterogeneity among pancreatic cancer types and patients. Consequently, many generalised therapies targeting only selected mutations or pathways that are present in a subset of carcinomas are often not effective. Therefore, genome-wide studies are crucial for understanding the contribution of variable genetic mutations to the carcinogenesis, progression and metastasis of pancreatic cancer in order to offer a personalised medicine approach by providing implications for further clinical achievements and drug development [21,38]. Unfortunately, the molecular characterisation in pancreatic cancer is not yet standard in clinical care, but great efforts are being made to use multiple insights provided by recent NGS-based studies in clinical trials [39]. According to Felsenstein et al., genetic alterations may be used to develop diagnostic markers for early detection, disease progression and novel targeted therapies [25]. As an example, modern screening methods, such as the analysis of pancreatic cysts and duodenal fluid, as well as circulating tumour DNA (ctDNA), incorporate the detection of molecular alterations. Mutations in KRAS or GNAS, which are present in more than 96% of patients with a mucin-producing premalignant neoplastic cyst, can distinguish harmful precursor lesions from benign cysts (e.g., serous cystadenomas) [40,41]. Moreover, altered late driver genes such as TP53 and SMAD4 are even more accurate markers that differentiate cysts requiring urgent clinical intervention from those that can be safely monitored. Mutations in TP53 and SMAD4 are common events in duodenal fluids from patients with pancreatic ductal adenocarcinoma and may be used to differentiate them from control patients. This is possible thanks to the innovative digital NGS technique detecting even low-abundance mutations down to 0.1–1% mutation prevalence. According to Yu et al. [42], mutations of TP53 and SMAD4 genes are often detected in duodenal fluids from patients with pancreatic ductal adenocarcinoma. Interestingly, these mutations allow for distinguishing patients with pancreatic ductal adenocarcinoma from control patients with non-suspicious pancreata. The “liquid biopsy” relying on ctDNA sequencing collected from peripheral blood samples is a powerful technique in metastatic pancreatic cancer recognition that can be used to identify invasive cancers earlier and to monitor patients with recognized cancer. It may be especially useful in the prediction of cancer recurrence, even before a computed tomography scan, and is correlated with survival among metastatic and resected patients [43,44].

It cannot be denied that in the last years, the oncological approach has also been revolutionised by a growing personalisation of the treatment. According to Melisi et al., this precision oncology approach is possible thanks to NGS, which enables genomic profiling [45]. The authors identified genomic alterations in 68 pancreaticoduodenal cancer patients who failed standard treatments. The mutations were characterised based on the ESMO Scale of Clinical Actionability for molecular Targets (ESCAT), which classifies cancer patients who are likely to respond to precision medicines. According to ESCAT, at least one alteration ranking of tier I, II, III or IV was detected in 8, 1, 9 and 12 patients, respectively (44.1%). The most frequent alterations with clinical evidence of actionability (ESCAT tier I-III) affect genes of the RAF (10.3%), BRCA (5.9%) or FGFR pathways (5.9%). The results obtained in this study enabled the selection of patients receiving a molecularly targeted matched therapy. Inconveniently, this genomic profiling method may be used only when standard treatments fail and in young patients with a good performance status [39,45]. 

The significance of genomic studies is not to be overestimated, and the full characteristics of the pancreatic genome may have promising implications for further clinical significance and drug development. Therefore, not only do known genes need to be studied, but also novel mutated coding regions, as well as non-protein coding genomics locations, such as non-coding RNA [46] and regulatory regions [47], that are likely to have functional significance in carcinogenesis. Such mutations located in functionally important genomic structures may represent putative predictive markers at the early stage of disease, before the onset of symptoms has an essential role in treatment and survival rates. 

## 4. Risk of Metastasis and Genetic Background of the Process

Pancreatic cancer is an aggressive tumour with frequently occurring distant metastases that can appear in virtually every organ. The liver is usually the most affected, followed by the lungs, peritoneum and bones. Various factors are associated with a metastatic pancreas tumour in comparison to an early-stage pancreas tumour, among them non-coding RNAs, transcription factors, growth factors and oxygen conditions as well as mutations [24,48]). 

Several studies have aimed to characterise genomic instability in metastatic pancreatic cancer. Campbell et al. reported that an instability pattern in pancreatic metastatic cancer is different, e.g., from that in metastatic breast cancer [49]. The pancreatic cancer could be characterised by breakage–fusion–bridge cycles that predicate specific abnormalities of cell-cycle control, namely, dysregulation of the G1-to-S transition and an intact G2–M checkpoint [49].

Moreover, selected genes play a role in the increase in pancreatic cancer metastatic activity. For example, a *DPC4* expression metastatic pancreatic cancer was higher compared with early-stage cancers or pancreatic intraepithelial neoplasia. Loss of the DPC4 protein has been shown to correlate with a shortened survival in patients with resectable pancreatic cancers. Among matched primary and metastatic cancer tissues, the loss of MKK4 in immunolabeling was observed in 11% of primary cancers, but in as many as 37% of distant metastases, indicating that MKK4 loss correlates with metastasis formation. On the other hand, other proteins’ expression, such as p53, showed no correlation between metastases and the primary tumour in immunostaining. 

The differential gene expression in pancreatic cancer has been assessed with different methods, with the global genetic expression being most often assessed with an array expression. However, most of the studies compared a differential genetic expression between tumour and normal tissues, not assessing the genetic and molecular differences between early and metastatic pancreatic tumours [50]. One of the few studies comparing early cancer and metastases concluded that metastatic spread is not accompanied by reproducible changes in gene expression [51]. The studies on mice reported that 25 significant up- and 181 downregulated genes were differentially expressed when comparing the liver metastases with the primary tumour. That shows that on the expression level, some differences between primary and metastatic tumours can be delineated. Eight genes (*PAI-1*, *BNIP3l*, *VEGF*, *NSE*, *RGS4*, *HSP27*, *GADD45A*, *PTPN14*) were additionally validated in a semi-quantitative immunohistochemical analysis and revealed a positive correlation to the array data [52]. Further studies are needed, although they are difficult to perform, as pancreatic tumours are usually already advanced at the time of diagnosis.

Until now, a definite “metastatic cancer signature” could not be identified, although there are events that are generally associated with the metastatic potential of tumours. [49,53]. Interestingly, in pancreatic metastasis, the clonal diversity differs between metastatic sites, with metastases in the lung and liver being monoclonal and those in the peritoneum and diaphragm being polyclonal. These findings indicate that clonal diversity depends on the metastatic site [54] and suggest the role of other mechanisms, such as the microenvironment, in the malignity of pancreatic cancers [53].

## 5. Hereditary Pancreatic Cancer

Approximately 10–15% of all pancreatic cancers, both exocrine and neuroendocrine, are inherited. Between 10–20% of all mutations and related genetic syndromes, described in the next paragraph, can increase the possibility of developing pancreatic cancer.

Exocrine pancreatic cancer develops from exocrine cells that make digestive enzymes and secrete them into the small intestine. Between 90–95% of all pancreatic cancers are exocrine. Neuroendocrine pancreatic cancer arises from clusters of cells in the pancreas, called islets, that produce hormones. They are neoplasms that exhibit neuroendocrine phenotypes, such as the production of certain neuropeptides, large dense-core secretory vesicles and a lack of neural structures [55]. Hereditary pancreatic cancer can be divided into two categories: FPC (familial pancreatic cancer) and syndromes that do not meet the criteria to be classified as an FPC but contain mutations that may increase the risk of pancreatic cancer. FPC is a ductal adenocarcinoma pancreatic cancer that occurs in families. Patients can be identified with FPC when one or more first-degree relatives have ductal adenocarcinoma. If one first-degree family member is diagnosed with pancreatic cancer, the risk for other family-members is 4–6%. If two family members are diagnosed, the risk increases to 8–12%, and with over two positive family members, the risk is 17–32% [56].

The association of familial pancreatic cancer with other malignant neoplasms was also investigated using the “Swedish Family-Cancer Database”, which includes over 11.5 million individuals. The results of this study demonstrated an association between pancreatic cancer in parents and melanoma in offspring. An increased risk of cancers of the lung, breast, small intestine, colon, testes and cervix has also been observed in people with a family history of pancreatic cancer [57].

## 6. Pancreatic Cancer Associated with Cancer Syndrome

Most of the scientific reports indicate an association between the appearance of specific variants, genes or environmental factors as risk factors for pancreatic cancer. However, few of them show pancreatic cancer as a risk factor for other types of cancer.

In their study, Cote and al. assessed whether a family history of pancreas cancer is a risk factor for first-degree relatives for ten specific cancers, i.e., breast, colon, bladder, lungs, lymphoma, melanoma, ovary, pancreas, prostate head, and neck. A doubled risk of lymphoma and ovarian cancer was observed in first-degree relatives of individuals with pancreatic cancer [58]. 

Pancreatic cancer appears to be a genetically heterogeneous disease that is an integral alteration of many cancer syndromes. Several cancer syndromes predispose one to an increased risk of pancreatic cancer. These include hereditary pancreatitis (HP), hereditary non polyposis colorectal cancer (HNPCC) or Lynch syndrome (LS), hereditary breast and ovarian cancer (HBOC), familial atypical multiple mole melanoma syndrome (FAMMM), Peutz–Jeghers syndrome (PJS), Ataxia–telangiectasia (AT), familial adenomatous polyposis (FAP), familial pancreatic cancer, familiar breast cancer, von Hippel–Lindau syndrome (VHL), Li–Fraumeni syndrome (LFS) and Fanconi anaemia [59,60,61,62,63,64,65,66,67,68,69]. An additional group represents cancer syndromes associated with pancreatic neuroendocrine tumours (pNETs). Approximately 1% to 2 % of pancreatic neuroendocrine tumours are present in the context of hereditary disorders [70]. These include Multiple Endocrine Neoplasia type 1 (MEN1) (Wermer syndrome), von Hippel–Lindau disease, neurofibromatosis 1 (NF-1) and the tuberous sclerosis complex (TSC) [71]. Based on this, several potential candidate genes for PC have been proposed. In a study by Hu et al., a significant association was observed between pancreatic cancer and inherited germinal mutations in the *CDKN2A*, *TP53*, *MLH1*, *BRCA2*, *ATM* and *BRCA1* genes [72]. Further studies have identified other deleterious mutations in pancreatic cancer patients in genes *CHEK2*, *MUTYH/MYH, BARD1*, *MSH2*, *NBN, PALB2* and *PMS2* [73]. The best-known syndromes associated with PC are hereditary breast and ovarian cancer syndrome and familial atypical nevus and melanoma syndrome [73]. Mutations in the *BRCA1*, *BRCA2* and *CDKN2A* genes were found in 7.4% of PC patients [74]. The risk of PC in cancer syndromes is summarised in Table 2.

### 6.1. Hereditary Breast and Ovarian Cancer 

Hereditary breast and ovarian cancer is inherited as an autosomal dominant disorder characterised by increased risk of breast cancer, ovarian cancer and other cancers such as prostate cancer, melanoma and pancreatic cancer. The disease is associated with pathogenic or likely pathogenic variants in one of the two genes *BRCA1* or *BRCA2*. Their protein products are involved in DNA repair, recombination, transcription and tumour suppression [90]. Previous studies reported an increased risk of PC in both *BRCA1* (1–3%) [77,91] and *BRCA2* (2–7%) carriers [92,93] (Table 2). In particular, the risk appears to be evident in families of Ashkenazi Jewish descent, where *BRCA1/2* founder mutations were identified in 5.5% of Ashkenazi patients with PC [94]. Interestingly, in most studies, the risk of MS in BRCA2 carriers was higher than in *BRCA1* carriers [75]. Recent studies have also revealed genes other than BRC1/2 that have been identified as a causative factor in HBOC syndrome. These are mainly genes associated with an increased risk of breast and ovarian cancer, but certain genes also have a proven link to pancreatic cancer. These include the genes *PALB2, TP53*, *STK11, ATM, MLH1* and *MSH2*. Interestingly, they are also strongly associated with other cancer syndromes [95]. 

### 6.2. Familial Atypical Mole and Multiple Melanoma Syndrome 

Familial atypical multiple mole melanoma syndrome is an autosomal dominant inherited disorder characterised by multiple dysplastic nevi and melanoma [96]. Approximately 20–40% of patients with FAMMM have a germinal mutation in the *CDKN2A* gene, located on chromosome 9p21, but only about 1% have a mutation in another *CDK4* gene on chromosome 12q14 [97,98]. Some *CDKN2A* mutations have also been linked to malignancies other than FAMMM, particularly PC [97]. The association between FAMMM syndrome and PC is well documented. *CDNK2* mutations are associated with an increased pancreatic cancer incidence of up to 20% by age 75 [99]. The risk of PC in patients with FAMMM is estimated to be 13–22 times higher than in the general population and additionally increases to 38-fold in FAAMM subjects with *CDKN2A* mutations [100,101]. Interestingly, nearly 90% of sporadic PDAC harbour alterations in CDKN2 [49].

### 6.3. Lynch Syndrome

Lynch syndrome, also known as hereditary non polyposis colorectal cancer, is an autosomal dominant disease associated with an increased predisposition to colorectal, gastric and endometrial cancers [102]. LS is caused by germline mutations in mismatch repair genes (MMR), including genes encoding DNA reparation proteins: *MLH1*, *MSH2*, *MSH6*, *PMS2* and *EPCAM* gene, which lead to epigenetic silencing of *MSH2* [103]. The malfunctioning variants are referred to as *path_MLH1*, *path_MSH2*, *path_MSH6* and *path_PMS2* [104]. Mutations in the *MLH1* and *MSH2* genes have a greater impact on DNA repair; therefore, patients carrying these mutations have a significantly higher risk of developing cancer compared to patients with *MSH6* or *PMS2* mutations [103,105]. Kastrinos et al. demonstrated that patients from families with germline *MMR* gene mutations have a cumulative risk of pancreatic cancer ranging from 1.31% up to the age of 50 years to 3.68% up to age of 70 years [68]. This represents an 8.6-fold increase compared with the general population [68]. Similar results of the cumulative incidence of PC for *path_MMR* variants carriers were reported by Møller et al., who estimated a cumulative risk of 1.1% up to age 50, 3.9% up to 70 and 6.2% up to 75 years of age [88]. 

### 6.4. Peutz–Jeghers Syndrome

Peutz–Jeghers syndrome is a rare autosomal dominant disorder. PJS is most often caused by a germline mutation in the serine/threonine kinase 11 or liver kinase B1 STK11/LKB1 genes on chromosome 19p13.3 [106]. The *STK11/LKB1* is a tumour suppressor gene that plays an important role in regulating the cell cycle [107]. Mutations in *STK11* are identified in 50% to 80% of families with PJS. In other patients, PJS is probably the cause of de novo mutations [108]. It is characterised by hamartomatous gastrointestinal polyps and pigmented cutaneous and mucocutaneous macules [108]. Individuals with Peutz–Jeghers syndrome have an increased risk for developing colorectal, gastric system, breast, uterine, cervical, lung, ovarian, testicular and pancreatic cancers [109]. The literature has reported that the cumulative risk of developing PC ranges from 11 to 36% at the age of 70 years, which presents a 136-fold increase compared with the general population [61,78].

### 6.5. Li–Fraumeni Syndrome 

Li–Fraumeni syndrome is a rare hereditary autosomal dominant disorder due to germline pathogenic or likely pathogenic variants in the *TP53* gene. Approximately 7% to 20% of patients with germline *TP53* pathogenic variants have mutations derived de novo. LFS is characterised by the development of a wide spectrum of childhood and adult-onset malignancies. The main cancers are adrenocortical carcinomas, breast cancer, central nervous system tumours, osteosarcomas and soft-tissue sarcomas [110]. The lifetime risk of cancer in individuals with LFS approaches 75% in males and almost 100% in females [110,111]. LFS is also associated with an increased risk of several additional cancers, including leukaemia, lymphoma, gastrointestinal cancers, cancers of head and neck, kidney, skin, lung, larynx, ovary, thyroid, prostate, testis and pancreas. In addition to these cancers, patients with LFS have a 7.3-fold increased risk of PDAC [65].

### 6.6. Familial Adenomatous Polyposis 

Familial adenomatous polyposis syndrome is an autosomal dominant disease, predominantly caused by a germline mutation in the adenomatous polyposis coli (*APC*) gene on chromosome 5q21 [112]. It is a disease of high penetrance and variable expression [86]. Approximately 20–30% of FAP cases arise from de novo mutations, without a family history [113]. The mutation site in the *APC* gene determines the progression, clinical phenotypic variations (including Gardner and Turcot syndromes) and the risk of development of extracolonic manifestations of the disease [112,114]. FAP is characterised by hundreds to thousands of adenomatous polyps of the gastrointestinal mucosa [115]. FAP is also correlated with the incidence of other malignancies, mainly colorectal cancer, gastric, duodenal, hepatoblastoma and desmoid tumours [116]. The lifetime incidence of pancreatic cancer in patients with FAP is relatively low. The lifetime risk has been estimated at around 2%, but it is still four times higher than in the general population [69,87]. Furthermore, reports on the genetic link between the two disorders are limited and remain inconclusive. Hori et al. demonstrated sporadic *APC* mutations in pancreatic cancer [117], which has not been confirmed in other studies [118]. Another group of investigators hypothesised that *APC* mutation may play a role in the initiation of pancreatic cancer [119].

### 6.7. Ataxia–Telangiectasia 

Ataxia–telangiectasia is an autosomal recessive disorder characterised by neurological symptoms, such as cerebellar atrophy with progressive ataxia, and a marked predisposition to cancer [120]. Ataxia telangiectasia is due to mutations of the ATM gene, whose product is involved in DNA double-break repair [121]. In a study of 166 familial pancreatic cancer probands, 2.4% were carriers of the deleterious *ATM* variant [120]. Large-scale sequencing-based studies have identified up to 18% of *ATM* mutations in some cohorts of patients with pancreatic ductal adenocarcinoma [122,123,124]. The risk of pancreatic cancer among *ATM* pathogenic variant carriers has been estimated at 6.3% by age 70 (Table 2) [125].

### 6.8. Fanconi Anaemia

Fanconi anaemia is characterised by bone marrow failure, developmental abnormalities and an increased risk to develop malignancies. FA is a predominantly autosomal recessive disease. Nineteen FA-related genes (FANC) whose products suppress interstrand crosslink sensitivity have been described. FA patients harbour biallelic mutations in a particular FA gene, with the exceptions of *FANCB* (X-linked) and *FANCR*/*RAD51*, which are autosomal dominant [126]. Mutations in most FA genes lead to a chromosomal instability disorder. Several studies have reported FA gene mutations in pancreatic cancers [127,128,129,130]. Roberts et al. reported protein-truncating variants of AF (*BRCA2, PALB2, FANC, FANCG* and *FANCM*) in individuals with familial pancreatic cancer [84]. In another study, Rogers et al. showed that germline and somatic mutations in *FANCC* and *FANCG* may contribute to pancreatic cancer [128]. Mutations in the *FANCC* gene have been detected in two young-onset pancreatic cases, but none in *FANCG* [131].

### 6.9. Multiple Endocrine Neoplasia Type 1

Multiple endocrine neoplasia type 1 is a rare autosomal dominant and hereditary endocrine tumour syndrome with high penetrance. Familial MEN1is characterised by the combined occurrence of tumours of the parathyroid glands, endocrine pancreas, duodenum, anterior pituitary and, less commonly, of the stomach, adrenal gland, thymus and lungs [132]. MEN1 is caused by several germline mutations in the tumour suppressor *MEN1* gene that encodes the protein menin [133]. In a study by Ito et al. in patients with pNETs, *MEN-1* was detected in 10% of patients [134]. On the other hand, pancreatic endocrine tumours occur in 40–70% of MEN-1 patients [135]. MEN-1-associated pancreatic endocrine tumours are most commonly gastrinomas and non-functioning tumours [136]. Pancreatic endocrine tumours are the leading cause of death in MEN-1 patients [137].

### 6.10. Von Hippel–Lindau Syndrome

Von Hippel–Lindau syndrome is a hereditary autosomal dominant disease caused by mutations in the *VHL* gene located on chromosome 3. It is a disease with high penetration, early onset and a high prevalence of clinical symptoms. VHL is characterised by multi-organ tumours of the central nervous system, kidney, adrenal and reproductive organs and pancreas. The most common type of tumour in VHL is hemangioblastoma. Renal cell carcinoma (RCC) and endolymphatic and pancreatic neuroendocrine tumours are also diagnosed in individuals with VHL [64,138]. Approximately 35–70% of patients with VHL have pancreatic lesions, and 5–17% develop neuroendocrine tumours [82,139]. In a study by Hammel et al., pancreatic disorders were found in 77% of patients with VHL. This included true cysts (91.1%), serous cystadenomas (12.3%), neuroendocrine tumours (12.3%) or combined lesions (11.5%). Interestingly, in 7.6% of patients, the pancreas was the only organ affected [140]. Reich et al. reported an age-dependent cumulative probability of pancreatic neuroendocrine neoplasm of 10 % up to 40 and 26% up to 60 years of age [141].

### 6.11. Neurofibromatosis 1

Neurofibromatosis 1, also known as von Recklinghausen disease, is an autosomal dominant disorder characterised mainly by neurocutaneous manifestations [142]. It is caused by a mutation in the *NF-1* gene, which encodes the neurofibromin protein, leading to uncontrolled cell proliferation and the development of some tumours [143]. Gastrointestinal involvement is rare but is associated with a significant risk of malignancy. To date, there are only a few case reports linking NF-1 to pancreatic neuroendocrine tumours [70,144,145].

### 6.12. Tuberous Sclerosis (Bourneville’s Disease)

Tuberous sclerosis is an autosomal dominant neurocutaneous multisystem disorder. The disease is due to mutations in one of two genes—*TSC1* located on chromosome 9q34 encoding for the protein hamartin, or *TSC2* located on chromosome 16p13.3 encoding for the protein tuberin [146,147]. Among familial cases of TSC, about half show an association with the *TSC1* gene and half with the *TSC2* gene [148]. Pancreatic endocrine tumours have been reported in a small percentage of patients with TSC [149,150,151].

As presented above, most of the studies provide evidence for the contribution of genetic background in the development of cancer syndromes and related PC. Considering this evidence, it seems reasonable to include individuals with cancer syndrome in screening. The aim of these screenings would be to identify pre-cancerous lesions, which would enable actions to be taken to prevent the development of cancer or to detect cancer at an early, still treatable stage.

## 7. Targeted Therapy Overview and Future Perspectives

Though immunotherapy, microbiological therapy and targeted therapy (examples in Table 3 below) remain the most promising treatment options for pancreatic cancer at the moment, we should not forget about traditional methods, such as surgery (often combined with neoadjuvant therapy), radiation and chemotherapy (including adjuvant therapy) [152,153]. Further, palliative care plays an important role in many PC cases, as this is still a tumour with a high mortality rate [153]. Chemotherapy of pancreatic cancer depends on the stage and previous/planned treatment; MFOLFIRINOX (modified leucovorin, 5-fluorouracil, irinotecan, oxaliplatin) or gemcitabine with capecitabine are often used for 6 months after radical resection [153]. There is currently no immunotherapy approved for PC treatment, as it is considered to be a less immunogenic cancer; however, it may be used in combination with chemotherapy, chemoradiotherapy, vaccines and cytokine antagonism [153]. In recent years, research on modern therapies is becoming more and more extensive, and it is probable that new methods will be combined with traditional methods of treating PC [152,153].

When it comes to developing new therapies, there is hope in organoid models that can simulate the characteristics of pancreatic tumours [154]. This is a rapidly evolving preclinical model that allows for studying the characteristics of pancreatic cancer and leads to an improvement in patient outcomes [154].

Currently, the standards for first-line treatment in PDAC are FFX and GnP regimens. Unfortunately, in most cases, first-line therapy is not successful, but there are still not enough RTCs. It is suggested that FFX followed by GnP or GnP followed by FFX is the most feasible or beneficial approach [155]. The high incidence of germline mutations in PDAC patients makes targeted therapy tempting [155].

Given the frequent instability of the genome, which refers to a range of genetic alterations, from point mutations to chromosomal rearrangements caused by damage to the DNA repair mechanisms, various interventions to accumulate double-strand breaks and to induce apoptosis have been investigated in pancreatic tumours, the most effective of which is treating patients with inhibitors of poly-(adenosine diphosphate ribose) polymerase (PARPi for short) [156,157]. The European Medicines Agency registration allows for the administration of PARPi as a maintenance therapy in patients with pancreatic cancer, in whom a germinal mutation in *BRCA1/2* was detected, and disease progression was not observed for a minimum of 16 weeks after first-line treatment with platinum therapy. Olaparib has been also investigated in clinical trials as a monotherapy [29]. Olaparib proved to be an effective maintenance therapy for BRCA germline patients and prolonged progression-free survival [29]. There are numerous indications that it is possible, after advanced genetic testing, to expand the group of patients receiving PARPi [158]. The results of clinical trials in ovarian cancer, in which PARPi have been used for a long time, demonstrated the effectiveness of PARPi also in patients with somatic changes in the *BRCA1/2* genes, as well as with characteristic changes in the genome associated with HRD or referred to as genomic instability. Moreover, the latest registration of PARPi, called Niraparib, assumes that this drug is administered to patients with ovarian tumours, regardless of their mutational status [158]. 

With the development and rising accessibility of the omics diagnostic methods and data analysis, we can better characterise the genetics of pancreatic cancer, although the relevance of genetic markers and signatures still needs to be understood [159,160]. High-throughput methods have revealed other non-germline genetic variants involved in PDAC pathogenesis. The in silico analysis of expression data has revealed new therapeutic perspectives, such as bleomycin, as potential therapeutic agents for PDAC [161], although other non-DNA biomarkers can characterise PDAC on the molecular level and the response to therapy [162]. Although the NTRK pathogenic variant is rare with PDAC, such tumours can be treated with larotrectinib [163], and first reports on such cases have been published [164].

Due to the short life expectancy of patients and the time-consuming nature of such analyses, there have been few studies to date extending the indication for the administration of PARPi in pancreatic cancer. 

Precision medicine is very promising for the treatment and early diagnosis of pancreatic cancer. It is also worth mentioning that there are three types of new-generation treatments for pancreatic cancer-associated mutations: nucleic acid drugs, small molecules and antibodies. Nucleic acid therapy has many advantages over traditional drugs, for example: the inhibition of previously inaccessible targets, faster and longer lasting responses than protein inhibition, less severe side effects and short development duration [165]. Since 90% of pancreatic cancers have a KRAS mutation, it is the most common target. Anti-tumour activity was detected in mice bearing this mutation. It was also tested in more than 1000 clinical trials, and few ASOs showed positive results. In addition, ASO targeting XIAP combined with gemcitabine showed some effect. Fifty-eight percent of patients that received this combination lived 6 months or longer [165].

There is no approved RNAi treatment for pancreatic cancer, although in clinical trials, Golan et al. targeted the most common gene—KRAS. Unfortunately, the clinical trials did not show promising results—in 83% of patients, no change accrued, and in 17%, there was very little effect, but the treatment is safe [165]. It can be implemented in combination with chemotherapy. Nishimura et al. presented a trial in which a fine-needle injection of RNA oligonucleotide-targeted CHST15 turned out to be safe [166]. Although there are many kinds of nucleic acid drugs, such as small interfering RNA, decoys, CpG oligodeoxynucleotides, microRNA, decoys and aptamers, which are very promising, unfortunately, only 30% of patients’ mutations can be treated with drugs, because it is very hard to inhibit targeted RNA in humans: *PALB2*, *ATM*, *HER2*, *MET*, *MLH1*, *MSH2*, *MSH6*, *PMS2*, *KRAS*, *BRCA1 and 2, PI3CA*, *PTEN*, *CDKN2A*, *BRAF* and *FGFR1* [167].

**Table 3 cancers-15-00779-t003:** Targeted therapies currently used and in clinical trials among pancreatic cancer patients.

Treatment	Gene and Indication	Side Effects	References
PARPi (e.g., Niraparib, Olaparib)	*BRCA1/2*—known or suspected BRCA gene mutation, no worsening after the treatment with platinum resistant agent; also targeting tumours with HRD, including mutations in: *PALB2, RAD51C, BRCA 1/2*	nausea, vomiting, diarrhoea or constipation, fatigue, anaemia, leukaemia, taste changes	[23,30,157,158,159]
Erlotinib	*EGFR*—advanced pancreatic cancer may be administered in combination with gemcitabine	neutropenia, febrile neutropenia, fatigue, nausea, infections, vomiting, mucositis	[168,169,170]
Larotrectinib and entrectinib	*NTRK*—pancreatic cancer with NTRK mutations, resistance to other treatment	fatigue, nausea, vomiting, constipation, weight gain, and diarrhoea	[163,164]
Everolimus	inhibitor of mammalian target of rapamycin (mTOR); mutations in TSC1 and TSC2 genes	stomatitis, infections, diarrhoea, peripheral edema, fatigue rash, anaemia, hypercholesterolemia, lymphopenia, elevated aspartate transaminase (AST), fasting hyperglycemia	[171]
Belzutifan	hypoxia-inducible factor-2 alpha (HIF-2α) inhibitor, used for the treatment of von Hippel–Lindau disease-associated cancers	anaemia, low oxygen levels—shortness of breath, headache, dizziness, tiredness; nausea; increased blood sugar (increased thirst or urination, dry mouth, fruity breath), abnormal kidney function tests	[172,173,174,175]
Palbociclib	*CDKN2A mutations*	nausea, diarrhoea, vomiting, decreased appetite, changes in taste, tiredness, numbness or tingling in arms, hands, legs, and feet, sores on the lips, mouth or throat	[176]
[experimental/clinical trials]	*TGFBR2*—known or suspected *TGFBR2* gene mutation or its pathway	unknown	[19,20,21,177,178]
[experimental/clinical trials]	*SMAD4 mutations*	unknown	[179,180,181]
[experimental/clinical trials]	*MLL3 mutations*	unknown	[182,183,184,185]

## 8. Drug Resistance Induced by Genetic and Genomic Alterations

Designing effective treatment regimens requires an understanding of all potential mechanisms of therapeutic resistance. Genetic and epigenetic changes in tumour cells may play a significant role in resistance to chemotherapy, radiotherapy and immunotherapy for PC.

Among the genetic alterations contributing to drug resistance, mutations in the tumour suppressor genes *TP53*, *SMAD4* and *CDKN2A,* and especially mutations in the KRAS oncogene, have been highlighted [19,186]. In pancreatic cancer, KRAS is the most frequently mutated oncogene [187]. The activation of mutations in KRAS, observed in more than 90 % of pancreatic ductal adenocarcinoma cases, are a critical factor involved in tumour initiation and progression. Studies have shown a positive correlation of KRAS with gemcitabine resistance [188]. Although the first therapies based on KRAS inhibitors have been introduced, they may only benefit a small group of patients. Sotorasib, adagrasib and other currently tested inhibitors mainly target the *KRAS* G12C mutation, and there are no approved drugs targeting KRAS G12D and G12V [189,190].

In addition, drug resistance may be acquired during therapy and may result from genetic mutations arising during the treatment process as well as adaptive mechanisms and natural clonal selection of tumour cells [191]. Among patients treated with KRAS G12C inhibitors, the resistance mechanism was detected in 45% of patients. Acquired KRAS gene alterations associated with the observed resistance to treatment included G12D/R/V/W, G13D, Q61H, R68S, H95D/Q/R, Y96C and amplifications in *KRAS* [192]. It is thought that secondary mutations may lead to an increase in the active KRAS protein bound to GTP and also prevent binding to the drug [193]. Awad et al. observed resistance to both adagrasib and sotarisib associated with R68S and Y96C mutations, and H95D, H95Q and H95R mutations resulted in resistance to adagrasib only. Interestingly, in patients with acquired resistance to adagrasib, additional pathogenic alterations occurring not in the *KRAS* gene itself, but they were identified in the RTK-RAS signalling pathways. These included mutations in the *NRAS*, *BRAF*, *MAP2K1*, *RET*, *NF1* and *PTEN* genes, and they also played a significant role in the development of treatment resistance [192]. Another aspect is resistance to radiotherapy, which has been found for mutations in *KRAS*, *TP53* and *CDKN2A* [194].

The occurrence of resistance may also result from epigenetic changes, such as the regulation of gene expression by small non-coding RNAs (miRNAs). The results from studies have shown that the expression of specific miRNAs in PC correlates with sensitivity or resistance to chemotherapy [195,196]. Gemcitabine-resistant cells were characterised by low expression levels of miR-101-3p and miRNA-124 [197,198]. Furthermore, gemcitabine treatment led to secondary changes in the levels of individual miRNAs (miRNA-17-5p, miRNA-21, miRNA-203), consequently leading to chemoresistance [199,200,201].

## 9. Conclusions

Advances in sequencing technology are now reducing the time needed to obtain results and opening new opportunities for patients. Moreover, advanced genomic testing also provides information on potential susceptibility to other available targeted therapies, giving patients the chance to receive effective treatment. The current state of knowledge regarding the genetics of pancreatic cancer and the emerging new possibilities of available treatment options reflects advances in genomic medicine and personalised medicine, providing new hopes for the near future.

## Figures and Tables

**Table 1 cancers-15-00779-t001:** Mutations in genes related to pancreatic cancer.

SNP ID or SNP Location	Germline/Somatic	References
SNPs located in the BRCA1 (location: 17q21.31) and BRCA2 gene (location: 13q13.1)	germline	[29]
Mutation in the PALB2 (parner and localiser of BRCA2) gene (location: 16p12.2)	germline	[30]
The KRAS G12D, KRAS G12V and KRAS G12R mutations in the KRAS gene (location: 12p12.1)	somatic	[31]
The KRAS G12R mutation in the KRAS gene (location: 12p12.1)	somatic	[32]
Mutations in the gene (location: 11q22.3) encoding the ATM serine/threonine kinase and the gene (location: 9p21.3) encoding the CDKN2A cyclin-dependent kinase inhibitor 2A	germline	[33]
Mutation in the GNAS gene (chromosome 20)	somatic	[34]
Mutation in the ATM gene	germline	[35]
Mutations in the MLH1, MSH2, MSH6 and PSM2 genes	germline	[36]
Mutations in the genes encoding the CPB1 and CPA1 carboxypeptidases	germline	[37]

**Table 2 cancers-15-00779-t002:** Risk of development of pancreatic cancer.

Cancer Syndrome	Gene(s)	Location	Mode of Inheritance	Relative Risk for PC Compared with the General Population	Cumulative Risk of PC (%)	References
HBOC	*BRCA2*	13q13.1	AD	3.5–10 fold	3–5% by age 70 2–7% lifetime risk	[60,75,76]
	*BRCA1*	17q21.31	AD	2.26–3 fold	1–3% lifetime risk	[75,76,77]
	*PALB2*	16p12.2	AD	2.37 fold	2–3% up to 80 years	[62]
PJS	*STK11/LKB1*	19p13.3	AD	76–132 fold	11–36% to age 65–70	[61,78,79]
FAMMM	*CDKN2A*	9p21.3	AD	13–38 fold	17% by age 70 up to 20% by the age of 75	[63,80,81]
VHL	*VHL*	16p13	AD	-	5–17%	[64,82]
LFS	*TP53*	17p13	AD	6–7.3 fold	-	[65,83]
FA	Fanconi anaemia complex genes: *FANCN (PALB2)* *FANNG* *FANCM* *FANCL* *FANCD1 (BRCA2)* *FANCC*	16p12.2 9p13.3 14q21.2 2p16.1 13q13.1 9q22.32	AR	-	-	[66,84]
HP	*PRSS1* *SPINK1*	7q35 5q32	AD	35–70 fold	7.2–18% lifetime risk	[67,85]
FAP	*APC*	5q21	AD	4.46 fold	2% lifetime risk	[69,86,87]
LS (HNPCC)	*MLH1,* *MSH2,* *MSH6,* *PMS2,* *EPCAM*	3p22.2 2p21 2p16.3 7p22.1 2p21	AD	5–8.6 fold	1.1% up to age 50 3.9% up to age 70 6.2% up to age 75	[68,88,89]
AT	*ATM*	11q22.3	AR	5.7–6.5 fold	0.08% by age 30 6.3% by age 70 9.5% by age 80	[71]
MEN1	*MEN1*	11q13	AD	-	Frequency PET: 80–100% (microscopic) 20–80% (clinical)	[71]
NF-1	*NF-1*	17q11.2	AD	-	Frequency PET: Uncommon (0–10%)	[71]
TSC	*TSC1* *TSC2*	9q34 16p13	AD	-	Frequency PET: Uncommon	[71]

AD—autosomal dominant; *APC*—adenomatous polyposis coli gene; AR—autosomal recessive; AT—ataxia–telangiectasia; *ATM*—ataxia–telangiectasia mutated gene, *BRCA1*—breast cancer 2 gene; *BRCA2*—breast cancer 2 gene; *CDKN2A*—cyclin-dependent kinase inhibitor 2 A gene; *EPCAM*—epithelial cell adhesion molecule gene; FA—Fanconi anemia, FAMMM—familial atypical multiple mole melanoma syndrome; FAP—familial adenomatous polyposis; HBOC—hereditary breast and ovarian cancer; HNPCC—hereditary non-polyposis colorectal cancer; HP—hereditary pancreatitis; LFS—Li–Fraumeni syndrome; LS—Lynch syndrome; MEN1—multiple endocrine neoplasia type 1; *MLH1*—MutL homolog 1 gene; *MSH2*—MutS protein homolog 2 gene; *MSH6*—MutS homolog 6 gene; NF-1—neurofibromatosis1 (von Recklinghausen’s disease); *PALB2*—partner and localiser of BRCA2 gene; PC—pancreatic cancer; PET—pancreatic endocrine tumour; PJS—Peutz–Jeghers syndrome; *PMS2*—PMS1 homolog 2, mismatch repair system component 2 gene; *PRSS1*—protease, Ser1 gene; *SPINK1*—serine peptidase inhibitor Kazal type 1 gene; *STK11*—serine/threonine kinase 11 gene; *TP53*—tumour protein 53 gene; TSC—tuberous sclerosis complex; *TSC1*—tuberous sclerosis complex 1 gene; *TSC2*—tuberous sclerosis complex 2 gene; VHL—von Hippel–Lindau syndrome; (-)—no data available.

## Data Availability

Not applicable.

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
