# Peer review of "Genetics, Genomics and Emerging Molecular Therapies of Pancreatic Cancer"

_cancers, 2023, doi:10.3390/cancers15030779_

Round 1

Reviewer 1 Report

The manuscript "Genetics, genomics and emerging molecular therapies of pancreatic cancer" by Liu et al. is an important review, with an aim to present the most recent data related to genetics, genomics, therapies and potential therapies for pancreatic cancer. In general, it is succinct and clear, however - it lacks details and is not entirely concise. Some paragraphs (listed below) require paraphrasing and clarification. 

Major issues:
1. The authors should decide whether the subject of their manuscript is pancreatic cancer in general, or with particularizing to the types/subtypes. The more valuable would be the latter, but given the occurrence and the fact that Cancers Special Issue refers to PDAC, it will be acceptable to focus on PDAC only (in that case the reviewer suggests modifying the title as well). If they decide, however, to include other exocrine, as well as neuroendocrine types of pancreatic cancer, some general remarks should be corrected, as not entirely accurate (e.g. line 205).
It will be very valuable to add a paragraph to concisely describe the types of that particular cancer and the differences between them.
2. Both parts, describing genetics and treatment, are not comprehensive enough for a review. The authors should avoid giving examples only (e.g. lines 112, 476, 497), but rather should list (perhaps a table) all known genes/mutations/treatment options for the particular pancreatic cancer types, similar to the only comprehensive part of the manuscript: the description of various cancer syndromes. The reviewer suggests adding e.g.: Blomstrand et al, BMC Cancer 2019; Parasido et al, Mol Cancer Res 2019; Shi et al, Front Oncol 2021.
3. The "treatment" part is very perfunctory. It need details on chemotherapy, immunotherapy and targeted therapy.

Minor issues:
4. Line 249 - it should be clarified what mutations and syndromes the authors are describing.
Line 253 - the description of the neuroendocrine pancreatic cancers is too basic, more details should be added.
Line 254-256 - this sentence needs clarification.
Line 172 - 174 - this sentence needs clarification, also with the regard to "control patients".
5. It is not clear whether the authors use the terms "syndrome" and "cancer syndrome" interchangeably in the text. These terms don't have the same meaning, so it needs explanation.
6. The reviewer suggest consulting the text with an English native speaker. 

Author Response

Dear Reviewer,

Thank you for your time spent on our manuscript. We’re more than happy to provide necessary improvements and corrections. Following your instructions, we’ve added a proper paragraph describing the main differences between existing tumour types. The manuscript has been supplemented with several new paragraphs according to your hints and suggestions, especially describing the types of pancreatic cancer and differences between them. The authors have eagerly added a paragraph about clinical trials and possible treatment options, also, how the genetic and genomic alterations affect the drug resistance phenomenon. Moreover, a new table has been created summarising existing targeted therapies with their potential side effects. We’ve also added some details on chemotherapy and other therapeutic approaches considered in pancreatic cancer.

After all the improvements, the manuscript has been read by two native speakers in order to make sure we haven’t omitted anything important. Thank you very much for all your time and effort.

Yours sincerely,

Authors

Reviewer 2 Report

This review manuscript comprehensively summarized the genetic and genomic alterations and potential molecular therapeutics in pancreatic adenocarcinoma. Generally, the manuscript was well written and documented. The summarized results, in particular the association between PC and the cancer-related syndromes are well organized. However, the manuscript is missing some important information, such as how the genetic and genomic alterations affect the drug-resistance. The “Targeted therapy overview and future perspectives” section is a little bit simple, need adding more information about some therapies in clinical trials.

Author Response

Dear Reviewer,

Thank you for your valuable consideration and for your time spent on our manuscript. We’re more than happy to provide necessary improvements and corrections. The manuscript has been supplemented with several new paragraphs according to your hints and suggestions. The authors have eagerly added a paragraph about clinical trials and how the genetic and genomic alterations affect the drug resistance phenomenon. Moreover, a new table has been created summarising existing targeted therapies with their potential side effects.

Thank you very much for all your time and effort.

Yours sincerely,

Authors

Round 2

Reviewer 2 Report

I suggest combining Table 1 -5 into one table, because they are showing the same thing. Make corresponding changes to other Tables.

Author Response

Dear Reviewer,

Thank you again for your valuable consideration and for your suggestions. We have made necessary improvements and corrections. The authors have eagerly merged the five tables into one. 

Thank you very much for all your time and effort.

Yours sincerely,

Authors
